# AdaMuon: Adaptive Muon Optimizer

## Abstract

We propose AdaMuon, a novel optimizer that combines element-wise adaptivity with orthogonal updates for large-scale neural network training. AdaMuon incorporates two tightly coupled mechanisms: (1) an element-wise second momentum estimator applied to orthogonalized update directions, and (2) a sign-stabilized orthogonal update, where the momentum is first sign-transformed before orthogonalization. These two components jointly enable variance-adaptive scaling while maintaining stable update geometry. In addition, AdaMuon employs an RMS-aligned rescaling strategy to match the root-mean-square update magnitude to Adam, allowing direct reuse of existing learning rate schedules without extra tuning. Experiments demonstrate that AdaMuon not only maintains stability but can surpass Adam by more than 40% training efficiency in large-scale scenarios.

## 1 Introduction

Optimization algorithms are a cornerstone of modern deep learning, directly shaping training dynamics and influencing both convergence speed and generalization performance. As model scales have grown to billions or even trillions of parameters (Brown et al., 2020; Chowdhery et al., 2023; Touvron et al., 2023; Liu et al., 2024b; Moonshot, 2025), optimizers face increasingly heterogeneous gradient landscapes and complex parameter geometries. Adaptive methods (Duchi et al., 2011; Tieleman, 2012; Loshchilov & Hutter, 2017), exemplified by Adam (Kingma & Ba, 2014), have become the de facto choice for large-scale training owing to their variance-based step size control and ease of tuning. However, over a decade after its introduction, the field is witnessing a growing demand for a new generation of optimizers that are better aligned with the computational and statistical challenges of training modern large foundation models.

Muon (Jordan et al., 2024) has been introduced as a representative of this emerging direction. It applies polar decomposition to transform raw momentum matrices into spectrally normalized, direction-only updates. This orthogonalization yields improved stability for large-scale two-dimensional parameter blocks such as transformer weight matrices (Liu et al., 2025; Shah et al., 2025; Chen et al., 2025; Tveit et al., 2025), and has already been deployed in ultra-scale training, including GLM-4.5 (Zeng et al., 2025) with 355B parameters and KIMI Moonshot (Moonshot, 2025) with over 1T parameters. It is widely regarded as a strong candidate for the next-generation optimizer that could potentially replace Adam in large-scale model training.

Building on this momentum, we aim to push the boundary of optimizers toward a paradigm that not only preserves the well-conditioned, geometry-aware updates of Muon but also adapts to the diverse statistical characteristics of individual coordinates—combining the efficiency of first-order methods with the robustness of second-order adaptivity. This motivation leads us to propose AdaMuon, an initial step toward this vision. AdaMuon incorporates element-wise variance adaptation into orthogonal updates through two tightly coupled mechanisms: an element-wise second momentum estimator applied after orthogonalization, and a sign-stabilized orthogonal update. Together with an RMS-aligned rescaling strategy for seamless integration with existing learning rate schedules, these components enable AdaMuon to unify matrix-level stability and coordinate-wise adaptivity, offering a principled path toward the next generation of large-scale optimizers. Experiments show that AdaMuon not only achieves stable convergence but also improves training efficiency by more than 40% compared to Adam in large-scale scenarios.

In the remaining of this paper, Sec. 2 reviews Muon and its extensions, along with the necessity and challenges of incorporating second momentum estimation; Sec. 3 presents the AdaMuon algorithm

in detail; and Sec. 4 demonstrates that AdaMuon consistently outperforms other methods across a variety of models and datasets. Overall, we show that AdaMuon is a versatile algorithm that scales effectively to large-scale models.

---

**Algorithm 1** The AdaMuon Optimizer

---

**Input:** Initial 2D-weights $\mathbf{W}_0 \in \mathbb{R}^{n \times m}$, loss function $\mathcal{L}$, learning rate $\eta$, weight decay $\lambda$, momentum $\beta$, Newton–Schulz steps $T$, small constant $\varepsilon$
**Output:** Updated weights $\mathbf{W}$
Initialize first momentum $\mathbf{M}_0 \leftarrow \mathbf{0}$, second momentum $\mathbf{V}_0 \leftarrow \mathbf{0}$
**for** each iteration $t = 1, 2, \ldots$ **do**
    Compute gradient: $\mathbf{G}_t = \nabla_{\mathbf{W}_t} \mathcal{L}(\mathbf{W}_t)$
    Update first momentum: $\mathbf{M}_t = \beta \cdot \mathbf{M}_{t-1} + \mathbf{G}_t$
    Compute sign-stabilized orthogonal direction: $\mathbf{O}_t = \text{Newton–Schulz}(\text{Sign}(\mathbf{M}_t), T)$
    Update second momentum: $\mathbf{V}_t = \beta \cdot \mathbf{V}_{t-1} + (1 - \beta) \cdot \mathbf{O}_t \odot \mathbf{O}_t$
    Apply second momentum update: $\hat{\mathbf{O}}_t = \mathbf{O}_t \oslash (\sqrt{\mathbf{V_t}} + \varepsilon \cdot \mathbf{1})$.
    RMS-aligned: $\gamma_t = 0.2 \cdot \sqrt{mn}/\|\hat{\mathbf{O}}_t\|_F$
    Update weights: $\mathbf{W}_{t+1} = \mathbf{W}_t - \eta(\gamma_t \hat{\mathbf{O}}_t + \lambda \mathbf{W}_t)$
**end for**

---

## 2 Preliminary

### 2.1 Muon Optimizer

Muon (Jordan et al., 2024) is a recently proposed optimization method designed for parameter tensors that can be represented as matrices. At iteration $t$, given the current weight matrix $\mathbf{W}_t \in \mathbb{R}^{n \times m}$ and its gradient $\mathbf{G}_t$, momentum $\beta$, and learning rate $\eta$, Muon updates parameters according to

$$\begin{aligned} \mathbf{M}_t &= \beta \mathbf{M}_{t-1} + \mathbf{G}_t, \\ \mathbf{O}_t &= \text{Newton–Schulz}(\mathbf{M}_t, T), \\ \mathbf{W}_{t+1} &= \mathbf{W}_t - \eta \mathbf{O}_t. \end{aligned} \tag{1}$$

where $\mathbf{M}_t$ denotes the momentum buffer at step $t$, initialized as a zero matrix for $t = 0$. The Newton–Schulz step (Bernstein & Newhouse, 2024) approximates the polar decomposition $\mathbf{O}_t$ of $\mathbf{M}_t$, which corresponds to $\mathbf{O}_t = \mathbf{U}\mathbf{V}^\mathsf{T}$ in the singular value decomposition (SVD) $\mathbf{M}_t = \mathbf{U}\mathbf{S}\mathbf{V}^\mathsf{T}$. $T$ in Eq. (1) denotes the number of iteration steps. This approximation avoids the high computational cost of a full SVD, and preserves the update direction while removing anisotropic scaling.

Specifically, Newton-Schulz step begins by normalizing the momentum matrix: $\mathbf{X}_0 = \frac{\mathbf{M}_t}{\|\mathbf{M}_t\|_F}$. Then, for each iteration $k$, $\mathbf{X}_k$ is updated from $\mathbf{X}_{k-1}$ as

$$\mathbf{X}_k = a\mathbf{X}_{k-1} + b(\mathbf{X}_{k-1}\mathbf{X}_{k-1}^\mathsf{T})\mathbf{X}_{k-1} + c(\mathbf{X}_{k-1}\mathbf{X}_{k-1}^\mathsf{T})^2\mathbf{X}_{k-1}, \tag{2}$$

where $a$, $b$, and $c$ are iteration coefficients, and $\mathbf{X}_T$ is the output after $T$ steps. Convergence requires tuning $a$, $b$, and $c$ so that the polynomial $f(x) = ax + bx^3 + cx^5$ has a fixed point close to 1. Following Jordan et al. (2024), we adopt $a = 3.4445$, $b = -4.7750$, $c = 2.0315$, and $T = 5$, which accelerate convergence for small singular values while maintaining stability.

### 2.2 Scaling Up for Muon

In practice for scaling up, Muon faces two practical challenges. First, the root-mean-square (RMS) magnitude of the weight matrices can become excessively large, exceeding the high-precision range of bf16, which is harmful. Second, Muon is applied only to two-dimensional parameters (e.g., weight matrices), while one-dimensional parameters (e.g., biases) are still optimized with Adam, necessitating separate learning rates and tuning complexity into existing pipelines.

To address address both issues, Liu et al. (2025) propose to control the RMS of Muon's weights via weight decay, and introducing a scaling factor $\gamma = 0.2\sqrt{\max(m, n)}$ to match the RMS norm of

Muon's updates to that of Adam, thereby enabling a unified learning rate schedule. The resulting update rule is

$$\mathbf{W}_{t+1} = \mathbf{W}_t - \eta \cdot (\gamma \mathbf{O}_t + \lambda \mathbf{W}_t), \tag{3}$$

where $\lambda$ is the weight decay coefficient. This simple yet effective adjustment stabilizes the training process while enabling Muon and Adam to share the same learning rate schedule, thereby allowing seamless integration into existing optimization pipelines.

### 2.3 SECOND-MOMENTUM IN MUON: NECESSITY AND CHALLENGES

#### 2.3.1 NECESSITY

Newton's method leverages curvature via the inverse Hessian to obtain locally optimal update directions, but for large-scale models this is impractical. Adaptive optimizers such as Adam and RMSProp approximate curvature through exponential moving averages of squared gradients, enabling variance-based step size adjustment. Similarly, Muon's polar decomposition inherently captures matrix-level second-order structure. From

$$\mathbf{O}_t = (\mathbf{M}_t \mathbf{M}_t^{\mathsf{T}})^{-1/2} \mathbf{M}_t = \mathbf{M}_t (\mathbf{M}_t^{\mathsf{T}} \mathbf{M}_t)^{-1/2}, \tag{4}$$

it follows that Muon, like Shampoo (Gupta et al., 2018), encodes row–column second-order interactions without explicitly storing full covariance matrices.

However, this global orthogonalization does not model the local variance structure of individual parameters. In real-world training, gradient distributions are often highly anisotropic at the element level—some entries exhibit large fluctuations, while others remain consistently stable. Applying a uniform update magnitude in such cases risks overshooting in noisy coordinates and under-updating informative but low-variance ones. This mismatch might slow convergence, reduce stability, and limit Muon's effectiveness in tasks with heterogeneous gradient statistics.

#### 2.3.2 CHALLENGES

To this end, we consider integrating element-wise second-momentum estimation into Muon. The most straightforward idea is to extend Muon by directly appending a second momentum accumulator, and then scaling the orthogonal update $\mathbf{O}_t$ with its variance estimate. However, this introduces two key design challenges:

- **First, determining which component to accumulate.** In Adam, the second momentum is naturally accumulated on the raw gradient $\mathbf{G}_t$, but in Muon the update direction $\mathbf{O}_t$ is obtained through polar decomposition of the momentum buffer $\mathbf{M}_t$. It remains unclear whether variance tracking should be applied to $\mathbf{G}_t$, $\mathbf{M}_t$, or $\mathbf{O}_t$, since each choice leads to different stability and normalization behaviors.

- **Second, achieving a normalization effect comparable to Adam.** In Adam, both the first and second moments are computed from the same gradient signal, making their ratio an effective per-element normalization of step size. In Muon, however, $\mathbf{M}_t$ may fluctuate substantially during the early and middle stages of training, causing the resulting $\mathbf{O}_t$ to also vary dramatically across coordinates. Consequently, no matter whether the second momentum is accumulated on $\mathbf{G}_t$, $\mathbf{M}_t$, or $\mathbf{O}_t$, the resulting statistics remain unstable and fail to provide a reliable normalization effect, even potentially amplifying noise.

A careful resolution of these challenges is essential to seamlessly integrate variance adaptivity into Muon without undermining its orthogonalization benefits or its inherent scale-invariant properties.

## 3 ADAMUON

To address the aforementioned challenges, in this section, we introduce **AdaMuon**, a novel optimizer that retains the advantages of Muon's orthogonal updates while automatically adjusting the scaling of individual elements. AdaMuon integrates two complementary mechanisms, **element-wise second momentum estimation and sign-stabilized orthogonal updates**, to simultaneously capture accurate second momentum statistics and normalize the update magnitude.

## 3.1 ELEMENT-WISE SECOND MOMENTUM ESTIMATION

To address the first challenge, we choose to accumulate the second-momentum term on $\mathbf{O}_t$, rather than on the raw gradient $\mathbf{G}_t$ or the momentum buffer $\mathbf{M}_t$. This design is principled for two reasons. First, the raw gradient $\mathbf{G}_t$ carries ill-conditioned scaling and directional noise that Muon's polar decomposition is specifically designed to eliminate, making it unsuitable for stable variance tracking. Second, while $\mathbf{M}_t$ provides a temporally smoothed gradient, it remains unstable at the element level during the early and middle stages of training. Moreover, since the final update direction is derived from $\mathbf{O}_t$, accumulating variance on $\mathbf{M}_t$ introduces a mismatch with the rescaling basis used in the update. In contrast, $\mathbf{O}_t$ provides a geometrically normalized and stable descent direction, offering a cleaner basis for variance estimation.

Formally, let $\mathbf{O}_t$ denote the orthogonalized update matrix obtained via Newton–Schulz iteration. We maintain an exponential moving average of its element-wise squared values:

$$\mathbf{V}_t = \beta \cdot \mathbf{V}_{t-1} + (1 - \beta) \cdot \mathbf{O}_t \odot \mathbf{O}_t, \tag{5}$$

where $\odot$ denotes Hadamard product. Here, $\mathbf{V}_t$ serves as the second momentum buffer for the orthogonalized update, analogous to the variance accumulator in Adam. The coefficient $\beta$ is inherited directly from Muon's momentum parameter, ensuring that AdaMuon does not introduce any additional hyper-parameters. The variance-normalized update direction is then obtained as

$$\widehat{\mathbf{O}}_t = \mathbf{O}_t \oslash (\sqrt{\mathbf{V}_t} + \varepsilon \cdot \mathbf{1}), \tag{6}$$

where $\oslash$ denotes element-wise division, $\sqrt{\mathbf{V}_t}$ represents the element-wise square root of the variance estimates, $\mathbf{1}$ is an all-ones matrix of the same shape as $\mathbf{O}_t$, and $\varepsilon$ is a small positive constant added to prevent the denominator from being 0. This step adaptively reweighs each element of the orthogonalized direction according to its estimated variance, suppressing noisy coordinates while preserving Muon's globally coherent update geometry.

## 3.2 STABILIZING ORTHOGONAL UPDATES VIA SIGN TRANSFORMATION

While the element-wise second-momentum estimator effectively captures variance in principle, it becomes less suitable during the early to mid stages of training. In this regime, gradients are unstable and may undergo large fluctuations, causing the momentum buffer $\mathbf{M}_t$ itself to vary substantially at the element level. Consequently, the polar decomposition produces orthogonal updates $\mathbf{O}_t$ that also fluctuate dramatically across coordinates. Accumulating such unstable $\mathbf{O}_t$ into a second momentum term fails to yield meaningful normalization and may even introduce adverse effects by amplifying noise rather than stabilizing it.

We hope that $\mathbf{O}_t$ can be both utilized in second momentum scaling and stable enough for variance-based normalization. To achieve this, we propose to first apply a transformation $f$ to $\mathbf{M}_t$, so that

$$\mathbf{O}_t = g(f(\mathbf{M}_t)), \ g(\cdot) = \mathrm{polar}(\cdot). \tag{7}$$

Recall that $g(c\mathbf{M}_t) = g(\mathbf{M}_t)$ for $\forall c > 0$, which indicates that $g$ is globally scale-invariant, preserving only the overall directional information of $\mathbf{M}_t$. However, $g$ alone does not stabilize element-wise fluctuations. $f$ is therefore designed to complement $g$: while $g$ enforces global directionality, $f$ operates element-wise to preserve coordinate-level orientation while mitigating volatility.

**Theorem 1** (Characterization of admissible element-wise transformations). *Let $f : \mathbb{R} \to \mathbb{R}$ be a function applied element-wise to $\mathbf{M}_t$ before the polar operator $g(\cdot) = \mathrm{polar}(\cdot)$. Suppose $f$ satisfies the following conditions:*

1. *(Scale invariance) $f(cx) = f(x)$ for all $x \in \mathbb{R}$ and $c > 0$.*

2. *(Sign consistency) For $x \neq 0$, $\mathrm{sign}(f(x)) = \mathrm{sign}(x)$.*

3. *(Odd symmetry) $f(-x) = -f(x)$.*

4. *(Bounded range) There exists $C < \infty$ such that $|f(x)| \leq C$ for all $x$.*

*Then $f$ must be of the form $f(x) = c \cdot \mathrm{sign}(x), \ c > 0$. Moreover, since $g$ is globally scale-invariant, the multiplicative constant $c$ is immaterial, and the unique canonical choice is $f(x) = \mathrm{sign}(x)$.*

The proof of Theorem 1 is shown in Appendix. A. Condition (1) ensures that per-coordinate magnitudes do not reintroduce instability when passed into $g$, aligning with $g$'s own scale-invariance at the global level. Conditions (2) and (3) constrain the directional behavior of $f$, while Condition (4) guarantees value stability. Taken together, $f(x) = \text{sign}(x)$, which satisfies all four desiderata. This yields a stabilized input to $g$, ensuring that $\mathbf{O}_t$ remains both orthogonal and robust enough for variance-based normalization.

## 3.3 RMS-ALIGNED RESCALING

To maintain compatibility with Adam's learning rate schedules, we scale the RMS norm of AdaMuon's update $\widehat{\mathbf{O}}_t$ (after second momentum estimation in Eq. (6)) to match Adam's empirical RMS value of $\approx 0.2$ (Liu et al., 2025), yielding

$$\gamma_t = \frac{0.2}{\text{RMS}(\widehat{\mathbf{O}}_t)} = \frac{0.2\sqrt{mn}}{\|\widehat{\mathbf{O}}_t\|_F}. \tag{8}$$

Finally, the rescaled update is applied to the parameters as

$$\mathbf{W}_{t+1} = \mathbf{W}_t - \eta\big(\gamma_t\widehat{\mathbf{O}}_t + \lambda\mathbf{W}_t\big), \tag{9}$$

The pseudo-code of AdaMuon is shown in Alg. 1. In addition, we provide further discussions on the omission of bias correction in the second momentum estimation (Appendix B), and the convergence analysis of AdaMuon (Appendix C).

## 4 EXPERIMENT

### 4.1 EXPERIMENTAL SETUP

**Baselines.** We compare AdaMuon against AdamW (Loshchilov & Hutter, 2017) and Muon (Jordan et al., 2024). Given Muon's strong performance in large-scale training, we consider these two baselines sufficient to demonstrate the effectiveness of AdaMuon. Muon here is specifically the Kimi-variant Liu et al. (2025). Since Muon operates only on matrices and applies a separate Adam optimizer for the remaining parameters, we set the learning rates of both optimizers to be the same. For AdamW, the first and second momentum coefficients are set to 0.9 and 0.95, respectively; Muon uses $\beta = 0.95$, and AdaMuon uses $\beta = 0.95$ and $\epsilon = 10^{-8}$. The weight decay $\lambda$ for these optimizers are set to 0.1.

**Model Architectures.** We evaluate AdaMuon on two representative model families: GPT-2 and Qwen2.5 models. All GPT-2 experiments are based on the nanoGPT (Karpathy, 2022) implementation of the GPT-2 architecture (Radford et al., 2019). We consider four scales—Small (125M), Medium (355M), Large (770M), and XL (1.5B). Following the default configurations in nanoGPT, we remove bias terms in all linear layers, use the GeLU activation function, and set the dropout rate to 0.0. Two modifications are applied: (1) replacing the original learned positional embedding (WPE) with Rotary Positional Embedding (RoPE) (Su et al., 2024), and (2) substituting the cosine learning rate schedule with the warmup-stable policy (i.e., the schdule that omits the decay phase in WSD (Hu et al., 2024) schedule entirely: after warmup, the learning rate is held constant). For Qwen2.5 (Qwen et al., 2025), we adopt the 1.5B and 7B dense model following the official architecture specifications.

**Datasets.** GPT-2 models are trained on the OpenWebText dataset (Gokaslan et al., 2019), which contains approximately 9B training tokens and 4.4M validation tokens, all tokenized with the standard GPT-2 tokenizer. For Qwen2.5 models, the dataset is collected from online corpora, with low-quality content removed through a combination of manually filtering rules and LLM-based quality assessment. Beyond general text, it contains high-quality code, math, and multilingual content.

**Training Details.** We mainly focus on the pre-training of the model. For GPT-2 experiments, all models are trained on approximately 50B (49.2B) training tokens for 100K steps, with a context length of 1024 and a warmup period of 2K steps. For Qwen2.5 experiments, 1.5B model is trained on 100B tokens with a context length of 8196 with 6e-4 learning rate, and 7B model is trained

on 235B tokens. For 1.5B model, we use 8.4B tokens for warmup, and 16.8B for 7B model. All the other hyper-parameters follow the respective official configurations. All other training settings, unless otherwise specified, are kept consistent across methods.

For GPT runs, we fix the effective batch size (EBS) to 60 sequences. By model size (Small to XL), we use the following pairs (batch size (bs), gradient accumulate steps (gas)) so that EBS = bs × gas = 60: (15, 4), (15, 4), (5, 12), (5, 12). For Qwen-1.5B, we use global batch size (gbs) = 512 with micro-batch size (mbs) = 2; for Qwen-7B, gbs = 1024, mbs = 2.

Table 1: Training efficiency of Muon and AdaMuon over AdamW. All improvements are computed relative to the AdamW baseline (49.2B training tokens).

| Method | LR | GPT-2 Small | | GPT-2 Medium | | GPT-2 Large | | GPT-2 XL | |
|--------|-----|-------|-----|-------|-----|-------|-----|-------|-----|
| | | Train | Val | Train | Val | Train | Val | Train | Val |
| Muon | $6 \times 10^{-4}$ | 25.46% | 21.57% | 22.10% | 23.34% | 28.56% | 24.71% | 27.32% | 28.24% |
| AdaMuon | | 34.25% | 33.73% | 30.63% | 31.96% | 37.33% | 31.34% | 35.82% | 31.04% |
| Muon | $1 \times 10^{-3}$ | 27.65% | 25.46% | 34.11% | 35.26% | 42.14% | 35.87% | 49.60% | 43.72% |
| AdaMuon | | 37.33% | 38.81% | 40.07% | 39.84% | 48.32% | 42.22% | 51.76% | 46.36% |

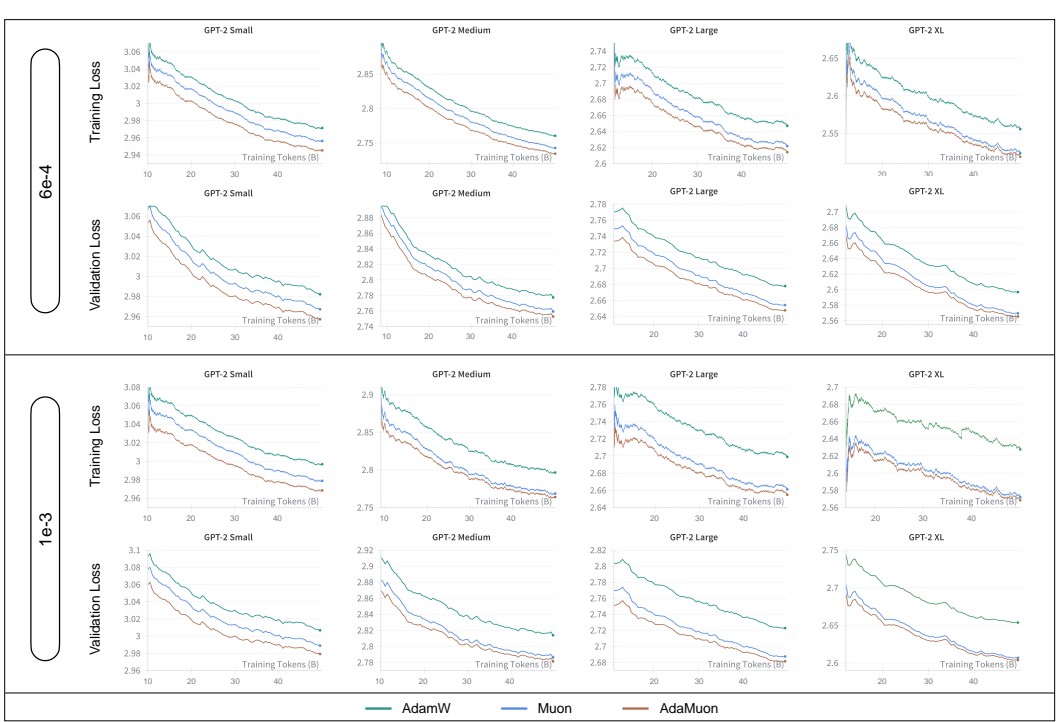

Figure 1: Training and validation loss comparisons of AdamW, Muon, and AdaMuon.

**Evaluation Metric.** We measure training efficiency improvement over Adam. For each method, we record the number of tokens required to reach the same training/validation loss achieved by Adam after training on $a$ tokens. If Muon or AdaMuon requires $b$ tokens to match this loss, the efficiency improvement is computed as $\frac{a-b}{a} \times 100\%$. This metric reflects the proportion of training tokens saved relative to Adam while achieving equivalent performance.

**Evaluation Benchmark.** We evaluate the Qwen model on a broad spectrum of 15 benchmarks. For English language understanding and reasoning, we adopt MMLU (Hendrycks et al., 2020), MMLU-Pro (Wang et al., 2024), TriviQA (Joshi et al., 2017), ARC-C (Clark et al., 2018), GPQA (Rein et al., 2024), OBQA (Mihaylov et al., 2018), HellaSwag (Zellers et al., 2019), WinoGrande

(Sakaguchi et al., 2021), PIQA (Bisk et al., 2020), and CommomsenseQA (Talmor et al., 2018). For code generation, we evaluate on the MBPP (Austin et al., 2021). For mathematical reasoning, we use the GSM8k (Cobbe et al., 2021). For Chinese language understanding and reasoning, we include CHID (Zheng et al., 2019), CMMLU (Li et al., 2023), and CEval (Huang et al., 2023).

## 4.2 RESULT

**GPT-2 Results.** Table 1 summarizes the training efficiency of Muon and AdaMuon under two learning-rate settings, and Fig. 1 illustrates the corresponding loss–token curves of AdamW, Muon and AdaMuon. Across all four GPT-2 scales, both Muon and AdaMuon deliver significant efficiency gains over AdamW, validating the benefit of geometry-preserving updates. Notably, AdaMuon consistently achieves the highest efficiency, regardless of model size or learning rate, demonstrating the effectiveness of integrating second-moment scaling with RMS-norm alignment. We also note that for GPT-2 XL with a learning rate of $1 \times 10^{-3}$, the efficiency gap is particularly large: in this regime, AdamW exhibits unstable training with multiple loss spikes, whereas Muon and AdaMuon remain stable, enabling them to reach the target loss substantially faster

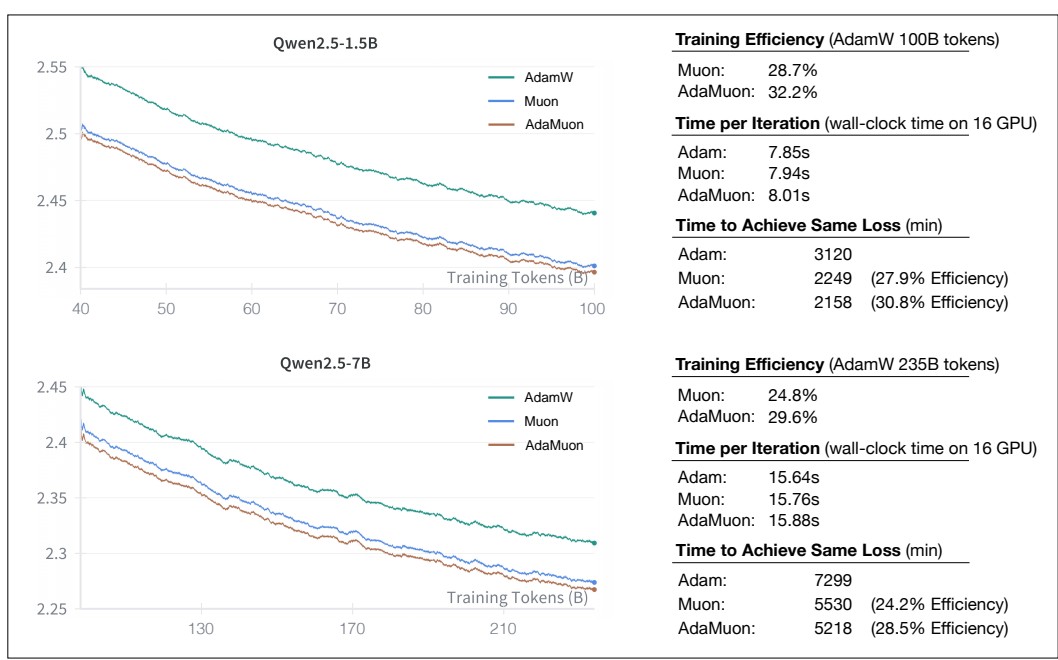

Figure 2: Results of AdamW, Muon, and AdaMuon when training Qwen2.5-1.5B and 7B dense models.

**Qwen2.5 Results.** Fig. 2 presents the training results of Qwen2.5. Consistent with the GPT-2 results, AdaMuon exhibits faster convergence than both Muon and AdamW across the entire training trajectory. In terms of time-to-target reduction, AdaMuon shortens the wall-clock time needed to reach the same loss by 30.8% compared to Adam and by 2.9% compared to Muon, translating into substantial savings in large-scale deployments. Moreover, we provide the evaluation results of Qwen2.5 after 100B-token training on 15 benchmark datasets. As reported in Table 2, models trained with AdaMuon consistently outperform those trained with Muon and Adam across all evaluation metrics, further confirming the effectiveness of our approach in both convergence efficiency and final task performance.

## 4.3 ABLATION STUDY

We conduct ablation experiments by selectively removing the sign operation and the second momentum term, with results summarized in Fig. 3. It can be observed that retaining only the sign

Table 2: Evaluation of Qwen2.5 models trained by AdamW, Muon and AdaMuon. Abbreviations: ARC-C = ARC-Challenge, WinoG = WinoGrande, ComQA = CommonsenseQA.

| Model | Optimizer | MMLU 5-shot | MMLU-Pro 5-shot | TriviaQA 5-shot | ARC-C 25-shot | GPQA 5-shot | OBQA 5-shot | HellaSwag 10-shot | WinoG 5-shot |
|---|---|---|---|---|---|---|---|---|---|
| | AdamW | 30.67 | 5.14 | 20.96 | 24.40 | 19.70 | 37.80 | 47.69 | 54.38 |
| 1.5B | Muon | **31.05** | 5.43 | **23.71** | **31.96** | 22.22 | 37.90 | 47.85 | **56.27** |
| | AdaMuon | 30.70 | **6.43** | 23.51 | 29.90 | **26.77** | **39.70** | **49.68** | 55.88 |
| | AdamW | 33.30 | 7.44 | 26.16 | 33.33 | 28.00 | 44.44 | 44.59 | 58.07 |
| 7B | Muon | 34.63 | 6.73 | **30.42** | 24.32 | 28.00 | **56.35** | 43.31 | **58.49** |
| | AdaMuon | **34.98** | **10.87** | 30.22 | **42.32** | **32.00** | 53.70 | **46.50** | 58.20 |

| Model | Optimizer | PIQA 5-shot | ComQA 7-shot | CHID 5-shot | CMMLU 5-shot | CEval 5-shot | MBPP 3-shot | GSM8k 4-shot | Avg. |
|---|---|---|---|---|---|---|---|---|---|
| | AdamW | 71.16 | 22.28 | 43.96 | **29.34** | **31.30** | 4.80 | 2.88 | 29.76 |
| 1.5B | Muon | **72.20** | 26.13 | 50.55 | 28.76 | 29.97 | 6.00 | 4.02 | 31.60 |
| | AdaMuon | 71.38 | **28.01** | **54.70** | 28.93 | 31.10 | **8.20** | **4.70** | **32.64** |
| | AdamW | 73.91 | 32.68 | 44.32 | 31.12 | 32.49 | 8.80 | 5.46 | 33.61 |
| 7B | Muon | **76.52** | **34.15** | **60.91** | **32.34** | 35.05 | 11.20 | **10.16** | 36.17 |
| | AdaMuon | 72.90 | 29.25 | 58.26 | 32.31 | **35.86** | **13.20** | 10.08 | **37.38** |

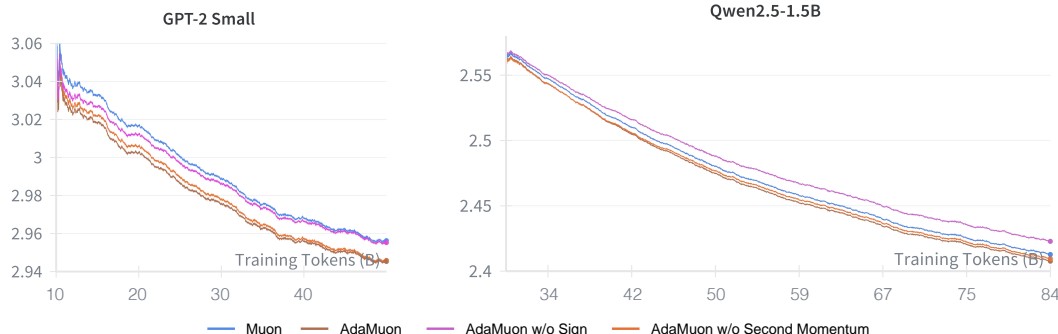

Figure 3: Ablation Study of AdaMuon. We present the training loss curve of GPT-2 Small and Qwen2.5-1.5B models.

operation still yields lower loss than the original Muon optimizer. This improvement stems from the fact that `sign` enhances the stability of the matrix obtained through polar decomposition, thereby stabilizing the update direction. In contrast, when only the second momentum term is added, the advantage disappears and Muon even outperforms this variant. This phenomenon is intuitive: directly accumulating second momentum can partly provide normalization, but since the orthogonal updates themselves remain unstable, the accumulated variance fails to stabilize training and may even harm optimization, leading to inferior loss reduction. Finally, when both `sign` and second momentum are retained, they complement each other: the former improves stability of the orthogonal update, while the latter provides effective element-wise normalization, resulting in the best overall performance.

### 4.4 TRAINING BEHAVIOR

In this subsection, we further analyze the training behavior of the three optimizers. All experiments are conducted on GPT-2 1.5B with a learning rate of $6 \times 10^{-4}$. We examine their gradient norms, parameter update norms, and the evolution of max attention logits across layers.

As shown in Fig. 4, the gradient norms of the three optimizers follow different trajectories. Muon exhibits a gradual upward drift, reflecting increasing gradient variance as training progresses. AdamW stabilizes quickly, maintaining a relatively flat trajectory with mild fluctuations, which indicates effective variance control. AdaMuon behaves similarly to AdamW but produces even smoother updates, highlighting its stronger capability in suppressing short-term oscillations.

The parameter norms further highlight their divergent behaviors. Both Muon and AdamW eventually converge to similar magnitudes, with Muon showing a sharper initial rise, consistent with its more aggressive gradient scaling. AdaMuon consistently produces smaller parameter norm, converging to a noticeably lower parameter norm. This restrained behavior helps stabilize training, although it may reduce the optimizer's ability to fully exploit the model's capacity compared to Muon or Adam.

Finally, the per-layer max attention logits reveal distinct adaptation patterns. In shallow layers (Layer 4), Muon steadily reduces the maximum attention logits, while AdamW maintains relatively high values, indicating stronger localized activations. In mid layers (Layer 25), all optimizers reduce logits, but AdaMuon stabilizes earlier, whereas Muon continues a slow decline. In deeper layers (Layer 45), AdamW sustains substantially higher logits than the others, while both Muon and AdaMuon settle at lower levels. These results suggest that AdamW encourages deeper specialization of attention heads, whereas Muon and AdaMuon promote a more uniform redistribution of attention across layers.

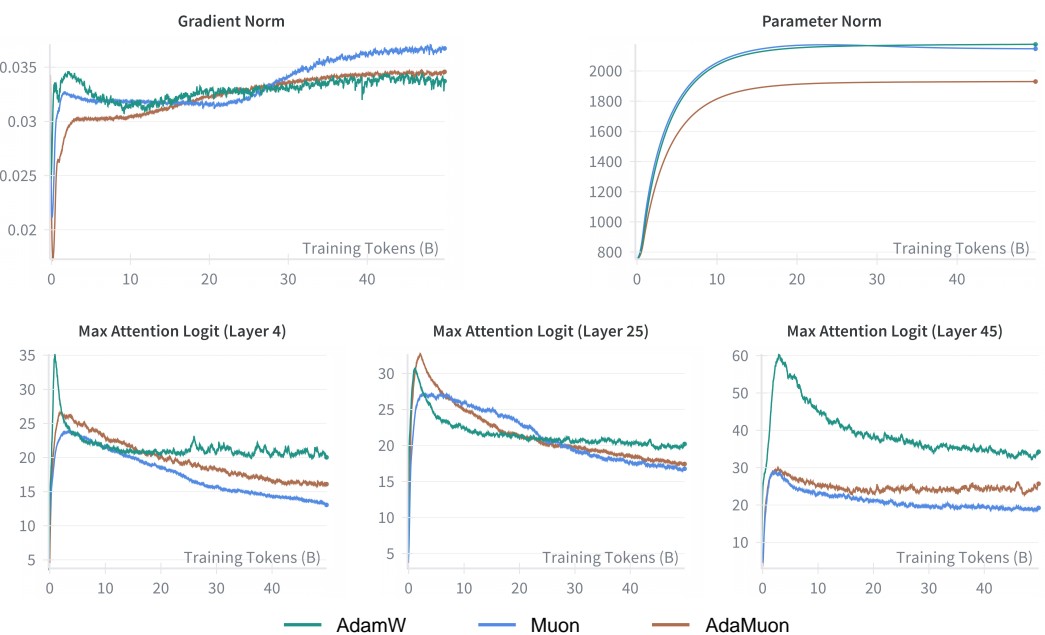

Figure 4: Training behavior of AdamW, Muon, and AdaMuon.

In addition, we note that prior work (Liu et al., 2025) reported that Muon tends to increase the maximum attention logits. Our findings differ: in the standard multi-head attention (MHA) setting, we do not observe excessively large logits under Muon. We argue that this discrepancy is not caused by the optimizer itself, but rather by the model architecture. Specifically, those studies employed variants with multi-latent attention (Liu et al., 2024a), where the structural design inherently amplifies logit magnitudes. By contrast, in conventional MHA layers, the effect of Muon on maximum logits remains moderate and comparable to other optimizers.

## 4.5 SENSITIVITY ANALYSIS

We evaluate AdaMuon's sensitivity to the first- and second-momentum coefficient $\beta$ on GPT-small with learning rate $6 \times 10^{-4}$. Varying $\beta$ over a standard range, the final-step training losses are shown in Table 3,

Table 3: Final training loss of varied $\beta$ on GPT2-small.

| $\beta$ | 0.8 | 0.9 | 0.95 | 0.99 |
|---|---|---|---|---|
| Loss | 2.96107 | 2.95735 | 2.95521 | 2.95487 |

which cluster tightly, indicating low sensitivity as long as $\beta$ remains in a reasonable band. Therefore, no special tuning is required—AdaMuon can directly reuse Muon's default $\beta$ (0.95).

## 5 CONCLUSION

This work presented AdaMuon, a new optimization paradigm that unifies the geometry-aware stability of Muon with the coordinate-wise adaptivity of variance-based scaling. By integrating a sign-stabilized orthogonal update, an element-wise second momentum estimator, and an RMS-aligned rescaling strategy, AdaMuon achieves both well-conditioned matrix-level updates and robust coordinate-wise adaptation—bridging the gap between efficiency and robustness in large-scale model training. Extensive experiments on GPT-2 and Qwen2.5 across multiple scales demonstrate that AdaMuon delivers consistent improvements over Adam and Muon, achieving up to **40%** gains in training efficiency with even superior final performance.

Looking forward, as model sizes and training demands continue to escalate, AdaMuon represents a promising step toward the next generation of optimizers that can scale efficiently while preserving stability in increasingly complex optimization landscapes. At the same time, we close with an open question to the community: how far are we from realizing a truly practical and scalable second-order optimizer for modern large-scale deep learning?

## 6 LIMITATION

AdaMuon also has some limitations. First, one additional second-moment buffer introduces more computation and memory. Second, there is minor per-step runtime overhead from Newton–Schulz and variance EMA. How to effectively tackle these drawbacks is also interesting, leaving for the future work.

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

# Contents

## A PROOF OF THEOREM 1

**Proof.** From Condition (2) and (3), $f$ must preserve orientation while being odd, hence $f(x)$ takes the same sign as $x$ and satisfies $f(-x) = -f(x)$. This constrains $f$ to functions of the form $f(x) = h(|x|) \cdot \text{sign}(x)$ with $h : [0, \infty) \to [0, \infty)$. From (1), for any $c > 0$, $h(c|x|) = h(|x|)$. Thus, $h$ is constant on $(0, \infty)$, i.e. $h(r) = \kappa$ for all $r > 0$. From (4), $\kappa$ must be finite. Therefore $f(x) = \kappa \cdot \text{sign}(x)$.

Finally, since $g$ is globally scale-invariant ($g(c\mathbf{M}_t) = g(\mathbf{M}_t)$), the multiplicative factor $\kappa$ vanishes in effect. Hence the canonical representative is $f(x) = \text{sign}(x)$.

## B WHY OMITTING BIAS CORRECTION IN ADAMUON'S SECOND MOMENTUM

Unlike Adam, AdaMuon does not perform bias correction on its second-momentum estimation $\mathbf{V}_t$ (line 8 in Algorithm 1). In Adam, the variance estimate is

$$\mathbf{V}_t = \beta \mathbf{V}_{t-1} + (1 - \beta)\, \mathbf{O}_t \odot \mathbf{O}_t, \tag{10}$$

which underestimates the true variance in early iterations. This bias must be corrected via

$$\widehat{\mathbf{V}}_t = \frac{\mathbf{V}_t}{1 - \beta^t} \tag{11}$$

to avoid shrinking the step size excessively. In AdaMuon, however, the variance-adaptive update

$$\widehat{\mathbf{O}}_t = \mathbf{O}_t \oslash (\mathbf{V}_t + \varepsilon \mathbf{1}) \tag{12}$$

is immediately followed by an RMS alignment step

$$\tilde{\mathbf{O}}_t = \gamma \, \widehat{\mathbf{O}}_t, \quad \gamma = \frac{0.2 \cdot \sqrt{mn}}{\|\widehat{\mathbf{O}}_t\|_F}, \tag{13}$$

which rescales the update to match a fixed target RMS magnitude (here, $0.2$, matching Adam's typical update norm). If $\mathbf{V}_t$ is biased by a constant factor $c$ (i.e., $\mathbf{V}_t \approx c \cdot \mathbf{V}_t^{\text{original}}$), then

$$\widehat{\mathbf{O}}_t \propto \frac{1}{\sqrt{c}}, \quad \gamma \propto \sqrt{c}, \tag{14}$$

and the scaling factor $\gamma$ cancels the bias exactly. Thus, the RMS-alignment step inherently removes any constant multiplicative bias in $\mathbf{V}_t$, making explicit bias correction unnecessary while keeping the update magnitude consistent with Adam.

By the way, AdaMuon does not apply bias correction to its first-momentum buffer $\mathbf{M}_t$. Any constant multiplicative factor $c$ in $\mathbf{M}_t$ has no effect on the normalized direction, because

$$\text{polar}(c\,\mathbf{M}_t) = \text{polar}(\mathbf{M}_t). \tag{15}$$

Thus, bias in the magnitude of $\mathbf{M}_t$ is inherently removed by the orthogonalization step, making explicit first-momentum bias correction unnecessary.

## C CONVERGENCE ANALYSIS

We analyze AdaMuon under standard smooth nonconvex assumptions. Weight decay is omitted ($\lambda = 0$) to focus on the core geometry; adding $\lambda > 0$ modifies constants in a routine way. All expectations are with respect to the algorithmic randomness and the stochastic gradients.

### C.1 ALGORITHMIC DEFINITIONS AND PRELIMINARIES

In the subsequent analysis, we will no longer use boldface for notation, but instead consistently adopt lightface for convenience. At iteration $t$, with parameter $W_t \in \mathbb{R}^{n \times m}$, let $G_t = \nabla \mathcal{L}(W_t)$ and let $\widehat{G}_t$ be a stochastic gradient such that $\mathbb{E}[\widehat{G}_t \mid W_t] = G_t$. AdaMuon forms

$$M_t = \beta M_{t-1} + \widehat{G}_t, \qquad S_t = \text{Sign}(M_t), \qquad O_t = \text{polar}(S_t),$$

and the element-wise EMA

$$V_t = \beta V_{t-1} + (1 - \beta)(O_t \odot O_t), \qquad V_0 = 0.$$

Define the normalized direction and RMS alignment factor

$$\widehat{O}_t \;=\; O_t \oslash \left(\sqrt{V_t} + \varepsilon \mathbf{1}\right), \qquad \gamma_t \;=\; \frac{c_{\mathrm{rms}}\sqrt{mn}}{\|\widehat{O}_t\|_F},$$

with a constant $c_{\mathrm{rms}} = 0.2$ (any fixed positive constant works in the analysis). The update is

$$W_{t+1} \;=\; W_t \;-\; \eta_t\,\gamma_t\,\widehat{O}_t.$$

Write $r = \min\{m, n\}$; then $\|O_t\|_2 = 1$ and $\|O_t\|_F^2 = r$.

**Assumption 1** (Smoothness). *$\mathcal{L}$ is $L$-smooth: for all $X, Y$, $\mathcal{L}(Y) \le \mathcal{L}(X) + \langle \nabla \mathcal{L}(X), Y - X \rangle + \frac{L}{2}\|Y - X\|_F^2$.*

**Assumption 2** (Unbiased stochastic gradients). $\mathbb{E}[\widehat{G}_t \mid W_t] = G_t$.

**Assumption 3** (Directional alignment after normalization). *There exists $\alpha \in (0, 1]$ such that, for all $t$, $\langle G_t, \widehat{O}_t \rangle \ge \alpha \|G_t\|_F$.*

**Discussion.** Assumption 3 expresses a positive cosine between the AdaMuon step direction and the true gradient. It accounts for (i) robustification by $\mathrm{Sign}(\cdot)$, (ii) orthogonal Procrustes alignment via the polar factor, and (iii) element-wise normalization by $(\sqrt{V_t} + \varepsilon)^{-1}$. In practice, the Newton–Schulz approximation error can be absorbed into a slightly smaller $\alpha$.

**Preliminaries on the preconditioner and RMS alignment**

**Lemma 1** (Preconditioner bounds and consequences). *Entrywise, $0 \le (V_t)_{ij} \le 1$ for all $t$. Hence*

$$\varepsilon \;\le\; \left(\sqrt{V_{t,ij}} + \varepsilon\right) \;\le\; 1 + \varepsilon, \qquad \frac{1}{1 + \varepsilon} \;\le\; \left(\sqrt{V_t} + \varepsilon\mathbf{1}\right)_{ij}^{-1} \;\le\; \frac{1}{\varepsilon}.$$

*Consequently,*

$$\|\widehat{O}_t\|_F \;=\; \left\| O_t \oslash \left(\sqrt{V_t} + \varepsilon\mathbf{1}\right) \right\|_F \;\le\; \frac{1}{\varepsilon}\|O_t\|_F \;=\; \frac{\sqrt{r}}{\varepsilon},$$

*and therefore*

$$\gamma_t \;=\; \frac{c_{\mathrm{rms}}\sqrt{mn}}{\|\widehat{O}_t\|_F} \;\ge\; c_{\mathrm{rms}}\sqrt{\frac{mn}{r}}\,\varepsilon \;=\; \underline{\gamma}.$$

*Proof.* Since $O_t \odot O_t \in [0, 1]^{n \times m}$ and $V_0 = 0$, the recursion $V_t = \beta V_{t-1} + (1 - \beta)(O_t \odot O_t)$ implies by induction $0 \le V_t \le \mathbf{1}$ entrywise. The stated bounds on $\sqrt{V_t} + \varepsilon$ and its inverse follow. Then $\|\widehat{O}_t\|_F \le \varepsilon^{-1}\|O_t\|_F = \sqrt{r}/\varepsilon$, implying the lower bound on $\gamma_t$. $\square$

**Lemma 2** (RMS alignment fixes the step norm). *For all $t$, $\|\gamma_t\,\widehat{O}_t\|_F = c_{\mathrm{rms}}\sqrt{mn}$.*

*Proof.* Immediate from the definition of $\gamma_t$. $\square$

## C.2 ONE-STEP PROGRESS

**Proposition 1** (One-step inequality). *Under Assumption 1,*

$$\mathcal{L}(W_{t+1}) \;\le\; \mathcal{L}(W_t) \;-\; \eta_t\,\gamma_t\,\langle G_t, \widehat{O}_t \rangle \;+\; \frac{L}{2}\,\eta_t^2\,\|\gamma_t\widehat{O}_t\|_F^2. \tag{16}$$

*If Assumption 3 holds, then*

$$\mathcal{L}(W_t) - \mathcal{L}(W_{t+1}) \;\ge\; \alpha\,\underline{\gamma}\,\eta_t\,\|G_t\|_F \;-\; \frac{L}{2}\,c_{\mathrm{rms}}^2\,mn\,\eta_t^2. \tag{17}$$

*Proof.* By $L$-smoothness with $Y = W_{t+1} = W_t - \eta_t \gamma_t \widehat{O}_t$,

$$\mathcal{L}(W_{t+1}) \leq \mathcal{L}(W_t) - \eta_t \gamma_t \langle \nabla \mathcal{L}(W_t), \widehat{O}_t \rangle + \frac{L}{2} \eta_t^2 \|\gamma_t \widehat{O}_t\|_F^2$$

Replace $\nabla \mathcal{L}(W_t)$ by $G_t$. Using Assumption 3 and Lemma 1, we obtain $-\eta_t \gamma_t \langle G_t, \widehat{O}_t \rangle \leq -\alpha \underline{\gamma} \eta_t \|G_t\|_F$. Using Lemma 2 gives $\frac{L}{2} \eta_t^2 \|\gamma_t \widehat{O}_t\|_F^2 = \frac{L}{2} c_{\mathrm{rms}}^2 mn \eta_t^2$. $\qquad\square$

Let $\Delta_0 = \mathcal{L}(W_1) - \mathcal{L}_\star$, where $\mathcal{L}_\star = \inf_W \mathcal{L}(W)$. Take expectations of Eq. (17) and sum over $t = 1, \dots, T$:

$$\alpha \underline{\gamma} \sum_{t=1}^T \eta_t \, \mathbb{E}\|G_t\|_F \ \leq \ \mathbb{E}[\mathcal{L}(W_1) - \mathcal{L}(W_{T+1})] \ + \ \frac{L}{2} c_{\mathrm{rms}}^2 mn \sum_{t=1}^T \eta_t^2 \ \leq \ \Delta_0 + K \sum_{t=1}^T \eta_t^2,$$

where $K = \frac{L}{2} c_{\mathrm{rms}}^2 mn$.

**Theorem 2** (Diminishing steps $\eta_t = \eta_0/\sqrt{t}$)**.** *With $\eta_t = \eta_0/\sqrt{t}$ and $\eta_0 > 0$,*

$$\min_{1 \leq t \leq T} \mathbb{E}\|\nabla \mathcal{L}(W_t)\|_F \ \leq \ \frac{\Delta_0}{\alpha \underline{\gamma} \sum_{t=1}^T \eta_t} \ + \ \frac{K \sum_{t=1}^T \eta_t^2}{\alpha \underline{\gamma} \sum_{t=1}^T \eta_t} \ = \ O\!\Big(\frac{\log T}{\sqrt{T}}\Big),$$

*since $\sum_{t=1}^T \eta_t \geq 2\eta_0(\sqrt{T} - 1)$ and $\sum_{t=1}^T \eta_t^2 \leq \eta_0^2(1 + \ln T)$.*

**Theorem 3** (Constant steps $\eta_t \equiv \eta$)**.** *With $\eta_t \equiv \eta > 0$,*

$$\frac{1}{T} \sum_{t=1}^T \mathbb{E}\|\nabla \mathcal{L}(W_t)\|_F \ \leq \ \frac{\Delta_0}{\alpha \underline{\gamma} T \eta} \ + \ \frac{K}{\alpha \underline{\gamma}} \eta \ \xrightarrow[T \to \infty]{} \ \frac{K}{\alpha \underline{\gamma}} \eta.$$

### C.3   PL-BASED CONVERGENCE

Assume the Polyak-Lojasiewicz (PL) condition:

**Assumption 4** (PL)**.** *There exists $\mu > 0$ such that $\frac{1}{2}\|G_t\|_F^2 \geq \mu\big(\mathcal{L}(W_t) - \mathcal{L}_\star\big)$ for all $t$.*

From Eq. (17) with constant step $\eta_t \equiv \eta$ and Lemma 1, we have

$$\mathcal{L}(W_{t+1}) - \mathcal{L}_\star \ \leq \ \mathcal{L}(W_t) - \mathcal{L}_\star \ - \ \alpha \underline{\gamma} \eta \|G_t\|_F \ + \ K \eta^2.$$

Using PL, $\|G_t\|_F \geq \sqrt{2\mu}\sqrt{\mathcal{L}(W_t) - \mathcal{L}_\star}$, hence

$$x_{t+1} \ \leq \ x_t - a\sqrt{x_t} + b, \quad x_t = \mathcal{L}(W_t) - \mathcal{L}_\star, \ a = \alpha \underline{\gamma} \eta \sqrt{2\mu}, \ b = K \eta^2. \tag{18}$$

**Lemma 3** (limit superior bound)**.** *For any $\varepsilon > 0$, there exists $T$ such that $x_t \leq (\frac{b}{a} + \varepsilon)^2$ for all $t \geq T$. In particular, $\limsup_{t \to \infty} x_t \ \leq \ \left(\frac{b}{a}\right)^2$.*

*Proof.* Fix $\varepsilon > 0$ and put $u = \frac{b}{a} + \varepsilon$. If $x_t \geq u^2$, then

$$a\sqrt{x_t} - b \ \geq \ a u - b \ = \ a\varepsilon,$$

so

$$x_{t+1} \ \leq \ x_t - \big(a\sqrt{x_t} - b\big) \geq a\varepsilon \ \leq \ x_t - a\varepsilon.$$

Thus whenever $x_t$ lies above $u^2$, it decreases by at least a fixed margin $a\varepsilon$. Hence this can only happen finitely many times: after some $T$, we must have $x_t < u^2$ for all $t \geq T$. Because $\varepsilon > 0$ is arbitrary, $\limsup_{t \to \infty} x_t \leq (b/a)^2$. $\qquad\square$

Based on Lemma 3, this recursion implies $x_t$ decreases monotonically until it enters the interval $[0, (b/a)^2]$ and then remains there. In particular,

$$\limsup_{t \to \infty} \mathbb{E}\big[\mathcal{L}(W_t) - \mathcal{L}_\star\big] \ \leq \ \frac{K^2}{2\mu \, \alpha^2 \, \underline{\gamma}^2} \, \eta^2.$$

Thus, under PL and the cosine-alignment Assumption 3, **AdaMuon converges to an** $O(\eta^2)$ **neighborhood.**

### C.4 CONVERGENCE SPEED (HITTING TIME TO THE $O(\eta^2)$ EIGHBORHOOD)

Recall the recursion $x_{t+1} \leq x_t - a\sqrt{x_t} + b$ with $a = \alpha\,\underline{\gamma}\,\eta\,\sqrt{2\mu}$ and $b = K\eta^2$. For any $\delta \in (0,1]$, define the target radius

$$x_\delta = (1+\delta)^2 \left(\frac{b}{a}\right)^2 = (1+\delta)^2 \frac{K^2}{2\mu\,\alpha^2\,\underline{\gamma}^2}\,\eta^2.$$

Choosing $\varepsilon = \delta\,\frac{b}{a}$ in Lemma 3, whenever $x_t \geq x_\delta$ we have $x_{t+1} \leq x_t - a\varepsilon = x_t - \delta\,b$, i.e. the loss gap decreases by at least $\delta\,b = \delta\,K\eta^2$ per iteration. Hence the number of iterations needed to enter the $(1+\delta)$-inflated $O(\eta^2)$ neighborhood satisfies

$$T_{\text{hit}}(\delta) \leq \max\left\{0,\ \left\lceil\frac{x_1 - x_\delta}{\delta\,b}\right\rceil\right\} = \mathcal{O}\left(\frac{1}{\delta\,\eta^2}\right),$$

where $x_1 = \mathcal{L}(W_1) - \mathcal{L}_\star$. In particular, to enter the radius $4\,(b/a)^2$ (i.e. $\delta = 1$) we have $T_{\text{hit}}(1) \leq (x_1 - 4(b/a)^2)/b = \mathcal{O}(1/\eta^2)$. Combining with $\limsup_{t\to\infty} x_t \leq (b/a)^2$ shows a standard trade-off: **larger $\eta$ reduces $T_{\text{hit}}$ as $\mathcal{O}(1/\eta^2)$ but enlarges the asymptotic neighborhood as $\mathcal{O}(\eta^2)$.**

**Optional: linear rate under a stronger alignment.** If, in addition to Assumption 3, there exists $\kappa > 0$ such that

$$\langle G_t, \widehat{O}_t \rangle \geq \kappa\,\|G_t\|_F^2 \quad \text{for all } t, \tag{19}$$

then the one-step bound becomes

$$\mathcal{L}(W_{t+1}) - \mathcal{L}_\star \leq \mathcal{L}(W_t) - \mathcal{L}_\star - \kappa\,\underline{\gamma}\,\eta\,\|G_t\|_F^2 + K\,\eta^2.$$

By PL, $\|G_t\|_F^2 \geq 2\mu\,(\mathcal{L}(W_t) - \mathcal{L}_\star)$, so for any $\eta \in \left(0, \eta_{\max}\right]$, $\eta_{\max} = \frac{1}{2\mu\kappa\underline{\gamma}}$,

$$\mathbb{E}\big[\mathcal{L}(W_{t+1}) - \mathcal{L}_\star\big] \leq (1-\rho)\,\mathbb{E}\big[\mathcal{L}(W_t) - \mathcal{L}_\star\big] + K\,\eta^2, \qquad \rho := 2\kappa\,\underline{\gamma}\,\mu\,\eta \in (0,1).$$

Therefore, AdaMuon enjoys *linear convergence* to an $\mathcal{O}(\eta^2)$ neighborhood:

$$\mathbb{E}\big[\mathcal{L}(W_t) - \mathcal{L}_\star\big] \leq (1-\rho)^t\big(\mathcal{L}(W_1) - \mathcal{L}_\star\big) + \frac{K}{\rho}\,\eta^2.$$

If the stronger directional condition $\langle G_t, \widehat{O}_t \rangle \geq \kappa\|G_t\|_F^2$ holds, then under PL we obtain a linear contraction $\mathbb{E}[\mathcal{L}(W_{t+1}) - \mathcal{L}_\star] \leq (1-\rho)\,\mathbb{E}[\mathcal{L}(W_t) - \mathcal{L}_\star] + K\eta^2$ with $\rho = 2\kappa\,\underline{\gamma}\,\mu\,\eta$, i.e. iteration complexity $T = \mathcal{O}\big(\frac{1}{\eta}\log\frac{1}{\varepsilon}\big)$ to reach an $O(\eta^2)$ neighborhood.

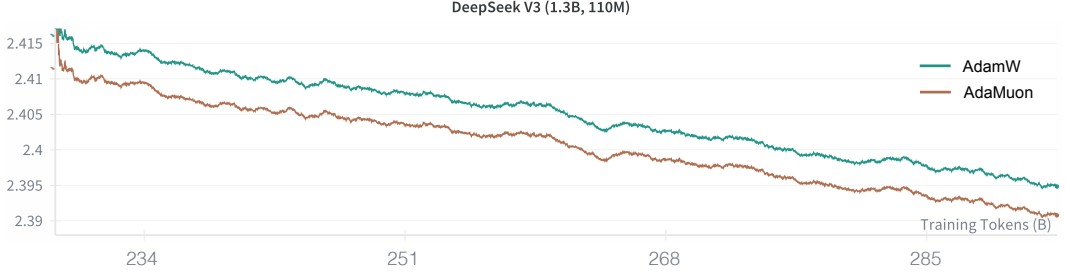

Figure 5: Results of AdamW and AdaMuon when training DeepSeek V3 models.

## D MORE EXPERIMENTS

### D.1 MOE ARCHITECTURE

To further demonstrate the effectiveness of AdaMuon, we evaluate it on a DeepSeek V3 (DSv3) (DeepSeek-AI, 2024) model. We modify the original configurations to obtain an MoE model of total 1.3B parameters, with 0.11B parameter activation. The training dataset used for DSv3 is the same as that used for training Qwen models. The batch size is set to 1024, with the sequence length 8192. The learning rate is set to $4.2\times10^{-4}$. We use 16.8B tokens for warmup, and continue train the model with total 300B tokens. The training loss curve are shown in the Fig. 5. Obviously, AdaMuon also performs well on MoE model.

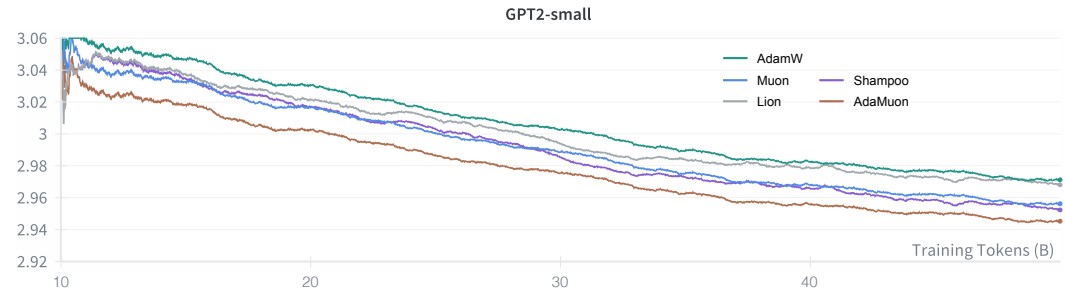

Figure 6: Results of AdamW, Muon, Shampoo, Lion, and AdaMuon when training GPT2-Small model.

## D.2 MORE BASELINES

In the main text we compared Adam (a canonical baseline) and Muon (the Kimi implementation regarded as a current strong optimizer). To broaden the comparison, we additionally evaluate Shampoo Gupta et al. (2018) and Lion Chen et al. (2023) under exactly the same training recipe on GPT2-Small with learning rate $6 \times 10^{-4}$. For both added baselines we adopt the authors' recommended default hyper-parameters without bespoke tuning, mirroring our fairness protocol in the main experiments. The resulting loss-vs-tokens and wall-clock-to-target curves are reported in Fig. 6. We observe that Shampoo is slightly stronger than Muon in this regime—intuitively reasonable because Muon can be viewed as Shampoo variant without explicit momentum accumulation. Nevertheless, AdaMuon retains a clear margin over Shampoo and all other baselines clearly, which further shows the effectiveness of AdaMuon.