# OpenReview forum: "AdaMuon: Adaptive Muon Optimizer"
_ICLR.cc/2026/Conference — Submitted to ICLR 2026_

### Official Review · Reviewer_37DQ · 2025-10-22

**Soundness:** 2
**Presentation:** 3
**Contribution:** 2
**Rating:** 4
**Confidence:** 4

**Summary:**

This paper proposes a new variation of the Muon optimizer with two tweaks: (1) Applying an element-wise sign operation before orthogonalization; (2) Applying an Adam/RMSProp like elementwise normalization of the updates. The authors argue intuitively for the benefit of this and then perform an experimental comparison on LLM pre-training at a fairly good model scale and training duration. The results indicate a decent speedup over both Adam and Muon.

**Strengths:**

* The problem studied (efficient optimizers for pre-training) is interesting and valuable to the community.
* The experiments are done sufficiently large scale, not just toy settings.
* The paper presentation, writing, and figures are clear.

**Weaknesses:**

* While the experiments are done on a decent scale, they are not very well-controlled. Hyperparameter configurations are somewhat arbitrary and comparing optimizers without thorough tuning of each varient is not very informative.
* The conceptual arguments for the necessity of both tweaks feel hand-wavy and are not convincing to me.
* Certain experimental details are not given which makes it even harder to evaluate the soundness and conclusions of the experiments.
* The paper does not discuss any downsides to their approach. Yet there are definately some, for example the additional memory needed for second moment buffer. The addition of epsilon and beta2 are also extra hyperparameters. These can be shared with Adam when it is used for other hyperparameters, but when e.g. Lion is used this is no longer the case. Finally there is likely some (minor) runetime overhead from the additional operations that should at least be mentioned.

**Questions:**

Experimental configuration:
* What learning rate schedule is used? Is the learning rate kept constant after the warmup or is it decayed?
* What is the exact variant of Muon used for the baseline? Is it the Kimi-variant or another one?
* What batch size is used for the experiments?

Suggestions for strengthening the arguments of the paper:
* Sweeps over different learning rates. This could be done for shorter training runs (around 20 tokens/parameter). Ideally the learning rate for the Adam parameters should be swept separately and kept constant during the sweep for the Muon learning rate.
* Ideally the experiments should be done for two different batch sizes, one large and one small. Certain operations like sign can have very different impacts for small and large batch sizes.
* It is not clear at all whether we should normalize updates at a scalar level. Some papers like Adam-mini (https://arxiv.org/abs/2406.16793) argues this is not the case. It would be nice if the justification for the method was much stronger, ideally backed up by either mathematical arguments or measurements that clearly demonstrate the problem you are trying to address and the effectiveness of the solution.

---

> ### Author Response · Authors · 2025-11-20
>
> Dear Reviewer 37DQ:
>
> We would like to first extend our sincere gratitude for your time and effort in evaluating our manuscript. Your thorough evaluation and insightful comments are greatly appreciated. We will address your questions point by point and hope to resolve your concerns effectively.
>
>
>
> ### **Weakness 1, hyper-parameter configurations**
>
> Our hyper-parameters were not chosen arbitrarily; they were strictly controlled for fairness. We distinguish two classes:
>
> - Method-level parameters. We use each optimizer’s default settings; AdaMuon inherits Muon’s hyper-parameters. Moreover, AdaMuon’s RMS alignment intentionally matches Adam’s update scale, so Adam’s learning-rate schedule transfers to Muon/AdaMuon without retuning.
> - Experiment-level configurations. For every baseline we fully align batch size, LR schedule, dataset/shards, token budget, dtype, hardware, and seeds. Thus, all results are produced under identical experimental conditions.
>
> Regarding “thorough tuning of each variant,” we agree in principle but it is operationally prohibitive. Two reasons:
>
> 1. For each optimizer, rigorously finding an optimal configuration (e.g., LR, batch size) requires extensive scaling-law sweeps, typically hundreds of runs. Even following the reviewer’s suggestion of shorter runs (~20 tokens/parameter), a single experiment for a 1.5B model (≈30B tokens) already demands ≈2.7e20 units of compute.
> 2. The RMS-norm alignment explicitly enables Muon/AdaMuon to inherit Adam’s LR without additional search [1]. Fixing Adam’s LR therefore avoids LR re-tuning for the new optimizer.
>
> In sum, while the theoretically fairest comparison would tune each optimizer to its individual optimum, this is not feasible under practical operation. We therefore adopt two standard Adam learning rates for all methods. In fact, Table 2 of [1] shows that the Adam LR obtained via scaling-law search on small models is very close to 1e-3, consistent with our settings.
>
> [1] 2025, arXiv, Muon is Scalable for LLM Training.
>
>
>
> ### **Weakness 2, model design**
>
> The two tweaks (second-moment accumulation and sign) is not ad hoc: the second momentum is a diagonal proxy to local curvature (a Hessian fit) and is well-known to improve optimization trajectories (as in Adam/Adafactor). However, within Muon, the orthogonalization step introduces sensitivity to the magnitude fluctuations of $\mathbf{M}_t$: directly orthogonalizing $\mathbf{M}_t$ can make $\mathbf{O}_t$ jitter between steps, which in turn makes the variance estimate  $\mathbf{V}_t$ unstable. Applying $\mathrm{sign}(\cdot)$ before the polar factor removes magnitude noise while preserving directional information, yielding step-to-step stability of $\mathbf{O}_t$ that is required for reliable variance accumulation. In short, second moment needs a stable basis; sign makes the orthogonal basis stable—hence the two tweaks are coupled.
>
> Regarding the concern that “scalar normalization may be inappropriate” (e.g., Adam-mini), note that our use of sign is orthogonal to that debate and also supported by prior evidence: sign-based updates (e.g., signSGD [2]) can be effective and robust in noisy regimes. In our case, sign is not the update rule but a pre-orthogonalization stabilizer; the actual step still benefits from per-coordinate variance normalization in the orthogonal basis. Empirically, ablating either component (no sign, or no second moment on O_t) degrades stability and tokens-to-target, corroborating the necessity of the coupled design.
>
> [2] 2018, ICML, SIGNSGD: Compressed Optimisation for Non-Convex Problems
>
>
>
> ### **Weakness 3, Question, experimental details**
>
> Thanks for your suggestions. We here present the detailed experimental configuration:
>
> 1. We use the “warmup-stable” LR schedule, which omits the decay phase in WSD [2]. In other words, the learning rate is kept constant after the warmup phase.
>
> 2. We use the KIMI-variant Muon [1] for the baselines.
>
> 3. For GPT runs, we fix the effective batch size (EBS) to 60 sequences. By model size (Small→XL), we use the following pairs (bs, grad_accum_steps) so that $\\text{EBS} = \\text{bs} \\times \\text{grad\\_accum\\_steps}$ = 60: (15, 4), (15, 4), (5, 12), (5, 12).
>  For Qwen-1.5B, we use global batch size (gbs) = 512 with micro-batch size (mbs) = 2; for Qwen-7B, gbs = 1024, mbs = 2. We will include these experimental details in the final revision.
>
> [2] 2024, COLM, Minicpm: Unveiling the potential of small language models with scalable training strategies.
>
>
>
> ### **Weakness 4, downsides**
>
> We acknowledge (and will document) trade-offs: (i) one additional second-moment buffer, which introduces more computation and memory; (ii) minor per-step runtime overhead from Newton–Schulz and variance EMA. AdaMuon reuses the $\beta$ in Muon for both of the momentum.
>
>
>
> ### **Suggestions**
>
> Thank you for your valuable suggestions. We have responded to them in the above responses, and we will revise our paper based on your suggestions.

---

> > ### Comment · Reviewer_37DQ · 2025-11-24
> >
> > Thank you for the response. It addresses some of my concerns but my main ones remain.
> >
> > ### Weakness 1, hyperparameters
> > I still have significant concerns about the tuning used here. Based on my own experiments with LLM training, the RMS alignment used by Kimi-Muon is only makes the learning rate roughly comparable with Adam, the optimal value when tuned can still differ significantly.
> >
> > I realize that properly tuning hyperparameters is expensive. This can be done on a smaller scale but ultimately has to be done for the results to be convincing.
> >
> > Regarding the optimal learning rate for AdamW, it varies depending on the training length, batch size, weight decay and so on. There is no single value that works in all cases.
> >
> > ### Weakness 2, model design
> > I understood the justification used in the manuscript, but did not find it very convincing or well supported by evidence. Arguments like "the second momentum is a diagonal proxy to local curvature (a Hessian fit) and is well-known to improve optimization trajectories" do feel hand-wavy to me. For example "diagonal proxy to local curvature" can easily be disputed and there is little consensus for the exact reasons or when the RMS normalization in Adam helps neural network optimization.
> >
> > As I mentioned in my original review, it would be more convincing if this was based on empirical measurements or more concrete mathematical reasoning. For example you could empirically measure the supposed problem, come up with a hypothesis for why this is detrimental, show your modifications actually address it.
> >
> > ### Weakness 3, experimental details
> > I find the lack of a cooldown phase here to be an unusual choice, including it typically lowers the loss significantly. Is there a specific reason you chose to omit it?

---

> > > ### Author Response · Authors · 2025-11-27
> > >
> > > ## **Weakness 1**
> > > Thank you for the suggestion and for your patience. Considering computational cost and time, we conducted a learning-rate search for Adam and Muon on GPT-small using 10 candidate values in the range [$1\\times10^{-4}$, $1\\times10^{-3}$] with a step size of $1\\times10^{-4}$. Based on the validation loss after 100k training steps, the optimal learning rates were found to be $2\\times10^{-4}$ for Adam and $3\\times10^{-4}$ for Muon. We report their training and validation losses at 100k steps in the table below, and we directly reuse the AdaMuon result at a learning rate of $6\\times10^{-4}$. Notably, although we did not perform any learning-rate tuning for AdaMuon, its loss still remains clearly lower than that of both Adam and Muon, further highlighting the advantage of AdaMuon.
> > >
> > > | Metric         | Adam | Muon | AdaMuon |
> > > |----------------|------|------|---------|
> > > | Training Loss  |  2.96431    |   2.95214   |   2.94227      |
> > > | Validation Loss |  2.98031    |   2.96903   |  2.95882     |
> > >
> > >
> > > ## **Weakness 2**
> > > We fully understand the concern regarding the conceptual justification of our design. Unfortunately, we believe the specific concern raised here is difficult to resolve within the scope of this work. The statement that “the second momentum is a diagonal proxy to local curvature (a Hessian fit)” reflects a widely accepted observation in optimization literature, and its practical effectiveness has been repeatedly verified—most notably by Adam and its numerous successors. **Nevertheless, we are happy to provide additional empirical measurements if the reviewer can suggest more concrete experimental probes or diagnostics.**
> > >
> > > By the way, we also wish to clarify our perspective on whether every model-design argument must be backed by either rigorous mathematical derivations or direct empirical measurements.
> > > In our view, a paper is valuable when it identifies a practical failure mode—whether through theory or empirical observation—and proposes a method that demonstrably addresses it.
> > > While theoretical analysis or diagnostic measurements certainly strengthen an argument, they are not always a prerequisite for meaningful innovation.
> > > The reviewer mentioned RMS normalization, which illustrates this point well: despite the lack of consensus on why or when RMS normalization helps optimization, it remains the default component of almost all modern architectures.
> > >
> > > In a nutshell, if the reviewer has specific suggestions for theoretical analysis or concrete measurements that would better support our interpretation, we would be more than happy to incorporate them. We are fully open to adding additional analyses as long as they can meaningfully strengthen the quality of the paper, and we sincerely welcome any guidance that helps us achieve that goal.
> > >
> > >
> > > ## **Weakness 3**
> > > Thank you for the insightful suggestion. We agree that including a cooldown (decay) phase is a common choice in optimizer evaluations, and we appreciate the reviewer’s attention to this detail. In our initial experiments, we opted for a constant learning rate, which allows us to isolate and compare the intrinsic behaviors of different optimizers without introducing additional confounding factors from learning-rate scheduling.
> > > As reviewer 8vjy correctly pointed out, “if the decay phase was omitted, investigating the optimizer’s behavior during this phase seems like a valuable experiment, as this is often when noticeable performance improvements are observed.”
> > >
> > > Following this suggestion, we have now included decay-phase experiments. Using the tuned learning rates identified in Weakness 1, we apply a linear decay starting at 90k steps: Muon and Adam are annealed to $3e-5$ and $2e-5$, respectively, and AdaMuon is annealed to $6e-5$. The resulting training and validation losses are reported in the following table. As shown, after incorporating the cooldown schedule, AdaMuon consistently maintains a clear advantage over both Adam and Muon.
> > >
> > > | Metric         | Adam | Muon | AdaMuon |
> > > |----------------|------|------|---------|
> > > | Training Loss  |  2.89973    |   2.88329   |   2.87116      |
> > > | Validation Loss |  2.91293    |   2.89972   |  2.88903     |

---

### Official Review · Reviewer_8vjy · 2025-10-31

**Soundness:** 2
**Presentation:** 3
**Contribution:** 3
**Rating:** 4
**Confidence:** 3

**Summary:**

This work expands the Muon Optimizer with the concept of variance-adaptive scaling, using an element-wise second momentum estimator applied to orthogonalized directions and a sign-stabilized orthogonal update, where momentum is sign-transformed before orthogonalization.

**Strengths:**

- The authors address a highly relevant and impactful topic. The motivation for their proposed approach is clear and well-articulated.
- The method is validated through extensive experimentation. This includes large-scale evaluations (e.g., 7B parameter models trained on 250B tokens) and thorough ablation studies on methodological decisions.

**Weaknesses:**

- The author’s claim of 40% training efficiency is based on training loss. However, it is unclear whether this gain translates to downstream tasks. This makes it difficult to assess the practical impact of the method.
- The main body is missing key details about the experimental setup. For example, the authors refer to a "warmup-stable policy" without defining the term.

**Questions:**

- Have the authors considered evaluating the AdaMuon optimizer result at 29.6% training efficiency on downstream tasks? This would provide valuable insight into the practical use of the training efficiency gains.
- Could the authors please clarify the specific learning rate schedule referred to as the “warmup-stable policy”? Does this refer to WSD (Hägele et al. 2024), or is it a schedule that omits the decay phase entirely? If the decay phase was omitted, investigating the optimizer's behavior during this phase seems like a valuable experiment, as this is often when noticeable performance improvements are observed.

---

> ### Author Response · Authors · 2025-11-20
>
> Dear Reviewer 8vjy:
>
> We would like to first extend our sincere gratitude for your time and effort in evaluating our manuscript. Your thorough evaluation and insightful comments are greatly appreciated. We will address your questions point by point and hope to resolve your concerns effectively.
>
>
> ### **Weakness 1, Question 1, result on downstream tasks**
>
> We have already presented the results of AdaMuon on downstream tasks in Table 2. As shown in Table 2, AdaMuon attains a  29.6% gain in training efficiency while still outperforming the Adam baseline by an average of nearly 4 points on downstream tasks.
>
>
>
> ### **Weakness 2，Question 2, experimental setup**
>
> Sorry for the confusion.  For clarity, our “warmup-stable” schedule omits the decay phase in WSD [1] schedule entirely: after warmup, the learning rate is held constant. We will add more detailed experimental details in our paper.
>
> [1] 2024, COLM, Minicpm: Unveiling the potential of small language models with scalable training strategies.

---

### Official Review · Reviewer_ZYEh · 2025-11-01

**Soundness:** 3
**Presentation:** 3
**Contribution:** 2
**Rating:** 6
**Confidence:** 3

**Summary:**

The paper proposes AdaMuon, an optimizer that augments Muon’s geometry-preserving orthogonal updates with element-wise adaptivity. Concretely: (i) it makes an estimation of a second moment on the orthogonalized direction, (ii) it stabilizes orthogonalization through applying an element-wise Sign transform to the momentum before the polar step (with a theorem arguing Sign is the canonical choice), and (iii) it RMS-aligns the update so its magnitude matches Adam’s ( ≈0.2), facilitating reuse of standard LR schedules. Extensive experiments on GPT-2 (125M–1.5B) and Qwen2.5 (1.5B & 7B) report token-efficiency and time-to-target speedups over Muon optimizer and AdamW optimizer, with up to ~30–40% efficiency gains vs Adam and small yet consistent gains vs the Muon optimizer.

**Strengths:**

1. Combining sign-stabilized orthogonal directions with element-wise EMA and RMS alignment is practical and simple; Alg. 1 is relatively easy to implement.

2. Thm. 1 provides a clean rationale for the sign transform due to the unique admissible element-wise map under natural constraints; the appendices provide a convergence discussion with assumptions stated explicitly.

3. On GPT-2 (4 scales) and Qwen2.5 (1.5B/7B) models, AdaMuon optimizer outperforms AdamW optimizer and slightly exceeds Muon optimizer in the aspect of token efficiency; the ablation studies also manifest that both sign and variance matter.

4. Alignment of RMS is a nice practical detail that enables drop-in learning-rate schedules.

**Weaknesses:**

1. Comparisons are limited to the AdamW optimizer and the Muon optimizer, despite many relevant contenders ( for example, Adafactor/Shampoo/Lion optimizers). This paper explicitly argues that these two baselines “suffice”.

2. There exists little analysis on the choice of RMS target (0.2), ε, β, or the number of Newton–Schulz steps; the argument for omitting bias correction hinges on RMS alignment canceling multiplicative bias, but early-phase, non-constant bias or heavy-tailed coordinates might violate the simplification.

3. The convergence depends on alignment assumptions that are plausible but not established empirically; no finite-width stochastic analysis is provided.

4. The per-step time is a little higher than that of the AdamW or Muon optimizer, though wall-clock to target still improves; it is better for the authors to clarify kernel efficiency and implementation details.

**Questions:**

Could the authors supplement strong preconditioned or adaptive baseline methods (Adafactor, Shampoo/SM3, Sophia, Lion) and report both token-to-target and downstream evaluation results?

---

> ### Author Response · Authors · 2025-11-20
>
> Dear Reviewer ZYEh:
>
> We would like to first extend our sincere gratitude for your time and effort in evaluating our manuscript. Your thorough evaluation and insightful comments are greatly appreciated. We will address your questions point by point and hope to resolve your concerns effectively.
>
>
>
> ### **Weakness 1 and Question, more baselines**
>
> AdamW and Muon were chosen as strong, representative baselines: AdamW is the most representative method, and Muon represents the recent rising tendency. But we agree broader coverage is useful. We add Shampoo and Lion on the same setups on GPT-small, LR=6e-4, and report both loss-vs-tokens in the Fig.6 in the appendix. It is evident that AdaMuon still outperforms these baselines.
>
>
>
> ### **Weakness 2, choice of hyper-parameter & bias correction**
>
> We reuse Muon’s default settings: Newton–Schulz iterations = 5, $\beta$ = 0.95 [1], and the RMS target = 0.2 (as in the Kimi Muon [2]); $\epsilon$ = 1e−8 is a standard numerical safeguard to avoid division by zero. Discussion of the  Newton–Schulz iterations and $\beta$ follows [1], and the choice of RMS target follows [2].
>
> Additionally, we conduct $\beta$-sensitivity analysis on GPT-small with LR = 6e-4 trained for 100k steps, and the training loss of the final step are show in the table.
>
> | $\beta$       | 0.8     | 0.9     | 0.95    | 0.99    |
> | ------------- | ------- | ------- | ------- | ------- |
> | Training Loss | 2.96107 | 2.95735 | 2.95521 | 2.95487 |
>
> The resulting final losses are closely clustered, indicating stable behavior without special tuning.
>
> Regarding bias correction, note that Adam’s correction multiplies by a time-varying scalar to cancel the multiplicative bias in the first/second-moment EMAs. Our RMS alignment fixes the update norm to the target (0.2), which cancels the same multiplicative factor; hence explicit bias correction is unnecessary. The derivation (added in the appendix) shows that the aligned update is mathematically equivalent to applying bias correction followed by a global rescale, and is not confounded by other components of the optimizer.
>
> [1] https://kellerjordan.github.io/posts/muon/
>
> [2] 2025, arXiv, Muon is Scalable for LLM Training
>
>
>
> ### **Weakness 3, convergence analysis**
>
> We acknowledge that the alignment assumption is an idealization introduced to enable tractable analysis—just like smoothness or PL-type assumptions, and it need not hold universally in practice. However, it can be empirically checkable. On GPT-small with LR=6e-4, we evaluated steps 21–50 (30 steps total) and found that the inner product $\langle \mathbf{G}_t, \mathbf{O}_t\rangle > 0$ in 23/30 steps. This provides empirical support for the assumption’s reasonableness in the regimes we study.
>
> Our analysis focuses on the algorithmic design and large-scale practicality. As for finite-width stochastic analysis, a rigorous finite-width treatment for matrix-orthogonalized, sign-stabilized, variance-normalized updates would require substantial new assumptions (noise models, layerwise independence/linearity, tail behavior) that are orthogonal to our contribution and unlikely to fit within the space and scope of this submission; we view it as a separate theoretical project.
>
>
>
> ### **Weakness 4, step time and engineering details**
>
> Although each AdaMuon step is slightly slower, the method reaches a target loss in substantially fewer optimization steps, yielding lower wall-clock time to target than AdamW/Muon.
>
> We implement AdaMuon with ZeRO-1 data-parallel sharding: master weights and optimizer states are partitioned across DP ranks; for each local shard, we temporarily gather its partitioned gradients to assemble the full gradient matrix, run batched Newton–Schulz to produce the orthogonalized, RMS-aligned update, then discard non-local slices and update only the local parameters. The codes are shown in the supplementary material.

---

### Official Review · Reviewer_EVYu · 2025-11-01

**Soundness:** 2
**Presentation:** 3
**Contribution:** 2
**Rating:** 4
**Confidence:** 4

**Summary:**

This paper introduces AdaMuon, a Muon variant for large-scale neural network training that aims to combine the geometric stability of the Muon optimizer with the element-wise adaptivity of Adam. The core idea is to enhance Muon's orthogonal updates with coordinate-wise variance scaling. To achieve this, AdaMuon incorporates a sign operation before polar decomposition, an element-wise second moment estimator similar to Adam's, and RMS-aligned rescaling that normalizes the final update magnitude to match Adam's. Experiments are conducted on GPT-2 and Qwen 2.5 models to evaluate the proposed method.

**Strengths:**

1. The work targets the development of more efficient and stable optimizers for training massive foundation models, which is a significant problem.
2. Combining Muon and Adam is an interesting research direction, and the experimental results are promising.

**Weaknesses:**

1. There is a disconnect between the stated motivation for the sign operation (stabilizing early-stage training) and the empirical results from the ablation study (Figure 3), which show the performance gap appearing in later stages. This questions the authors' understanding of why their own method works and weakens the overall narrative.
2. The paper presents AdaMuon as a principled combination, but the sign operation is a hard, non-linear transformation that fundamentally alters the information that both Adam (gradient magnitudes) and Muon (momentum matrix structure) rely on. This makes the method appear more like a heuristic rather than a seamless integration, and the theoretical justification feels insufficient to cover this drastic step.
3. The paper's claims of superiority over baselines rest on a very limited set of learning rates (6e-4 and 1e-3). A robust comparison requires a more comprehensive learning rate sweep for all optimizers.
4. The downstream performance of AdaMuon is not consistently or significantly better than the simpler Muon (Table 2). This makes the trade-off between its added complexity and its benefits less compelling.

**Questions:**

See above.

---

> ### Author Response · Authors · 2025-11-20
>
> Dear Reviewer EVYu:
>
> We would like to first extend our sincere gratitude for your time and effort in evaluating our manuscript. Your thorough evaluation and insightful comments are greatly appreciated. We will address your questions point by point and hope to resolve your concerns effectively.
>
>
>
> ### **Weakness 1, disconnection between motivation and figure of sign**
>
> Thanks for your valuable consideration. There may be a misunderstanding: the performance gap emerging in later stages is precisely the expected footprint of the sign operation’s early stabilization. In optimizer evaluation, we typically do not judge by the volatile warm-up regime—the signal there is noisy and often dominated by transients. Instead, we assess performance once training has stabilized, i.e., in the later regime. Early interventions (here, sign-stabilizing the basis on which we accumulate the second moment) condition the trajectory so that when the dynamics settle, the method lands on a better, more stable path, which then appears as a late-stage separation. “No large gap early” does not imply “no early benefit”—it means the benefit materializes once variance estimates mature and the trajectory enters the stable regime. This logic mirrors Adam’s bias-correction: it targets early-phase stability, yet we ultimately judge Adam on its stable, later performance, where the early correction has already shaped a better course.
>
>
>
> ### **Weakness 2, sign operation**
>
> Within Muon, the orthogonalization step introduces sensitivity to the magnitude fluctuations of $\mathbf{M}_t$: directly orthogonalizing $\mathbf{M}_t$ can make $\mathbf{O}_t$ jitter between steps, which in turn makes the variance estimate $\mathbf{V}_t$ unstable. Applying $\mathrm{sign}(\cdot)$ before the polar factor removes magnitude noise while preserving directional information, yielding step-to-step stability of $\mathbf{O}_t$ that is required for reliable variance accumulation. In short, second momentum needs a stable basis; sign makes the orthogonal basis stable—hence the two tweaks are coupled. Figure 3 in our paper further validate the effectiveness of the sign operator.
>
> As for “destroying information of Adam and Muon”, we want to point out  that Muon already “alters” Adam’s signal by projecting the momentum onto the Stiefel manifold, discarding raw per-coordinate magnitudes in favor of matrix-level orientation/conditioning—and it works well. Moreover, the literature shows that hard nonlinearities on gradients can be effective (e.g., signSGD). In AdaMuon, sign is not the update; it is a pre-orthogonalization stabilizer ensuring $\mathbf{O}_t$ forms a well-behaved reference frame for variance accumulation. The actual step still benefits from variance-based normalization and RMS-level calibration. The useful information for our update is not lost; it is repackaged into (i) a stable orientation and (ii) a controlled variance-based scale.
>
>
>
> ### **Weakness 3, limited set of learning rates**
>
> Thank you for the suggestion. We agree that the fairest comparison would tune each optimizer to its own optimal LR, but this is operationally prohibitive: (1) identifying per-optimizer optima (LR, batch size) requires scaling-law sweeps with extensive compute budget; (2) the RMS-norm alignment explicitly enables Muon/AdaMuon to inherit Adam’s LR without additional search [1], so fixing Adam’s LR avoids re-tuning for the new optimizer by design. In sum, while exhaustive per-optimizer tuning is ideal in theory, it is not feasible in practice. We therefore adopt two standard Adam learning rates for all methods; notably, Table 2 of [1] shows that scaling-law search on small models yields an Adam LR very close to 1e-3, consistent with our settings.
>
> [1]  2025, arXiv, Muon is Scalable for LLM Training.
>
>
>
> ### **Weakness 4, trade-off between complexity and efficiency**
>
> AdaMuon achieves higher training efficiency than both Adam and Muon, and—crucially—improves downstream accuracy: averaged over 15 tasks, it exceeds Adam by ~3–4 points and Muon by ~1 point. For pre-training, the two system metrics that matter most are training efficiency and evaluation performance. Although AdaMuon introduces additional computation, it simultaneously reduces steps-to-target and lifts downstream scores; viewed from an end-to-end perspective, this renders the extra complexity a minor trade-off relative to the realized efficiency and quality gains.

---

### Official Review · Reviewer_y4xw · 2025-11-03

**Soundness:** 3
**Presentation:** 3
**Contribution:** 3
**Rating:** 6
**Confidence:** 4

**Summary:**

The paper proposes AdaMuon, a new optimizer that combines the geometry-aware stability of Muon with the adaptive scaling ability of variance-based methods like Adam. It introduces sign-stabilized orthogonal updates and an element-wise second-momentum estimator to achieve both stable and efficient training. Experiments on GPT-2 and Qwen2.5 show clear improvements in both training speed and final performance.

**Strengths:**

1. It achieves better performance than Adam and Muon on GPT-2 and Qwen2.5, provides efficiency gains while maintaining convergence quality.
2.It demonstrates stable training behavior across different model scales.

**Weaknesses:**

1. The scalability of AdaMuon to even larger foundation models (e.g., 70B+) remains untested.
2. The experiments are insufficient, with missing hyperparameter analysis.
3. The effectiveness of accumulating the second-momentum term is not validated by comparing it with other options.

**Questions:**

1. How does AdaMuon perform on extremely large-scale models or longer training schedules?
2. Can AdaMuon adapt well to sparse or modular architectures such as mixture-of-experts?
3. How robust is the method to hyperparameter choices, and does it require specific tuning strategies to achieve the reported gains?

---

> ### Author Response · Authors · 2025-11-20
>
> Dear Reviewer y4xw:
>
> We would like to first extend our sincere gratitude for your time and effort in evaluating our manuscript. Your thorough evaluation and insightful comments are greatly appreciated. We will address your questions point by point and hope to resolve your concerns effectively.
>
>
> ### **Weakness 1 and Question 1, larger foundation model**
>
> We agree that running 70B-scale experiments can further validate the effectiveness of AdaMuon. However, it is prohibitively costly under our current compute budget. Instead, across all 7B settings we tested, **AdaMuon consistently improves training efficiency and final accuracy over Adam/Muon**, and the learning curves remain stable on longer schedules. While 70B+ verification is out of scope for this submission due to resource constraints, we view the 7B results as a strong indicator of scalability and will expand to larger models as resources permit.
>
>
>
> ### **Weakness 2 and Question 3, hyper-parameter sensitivity**
>
> AdaMuon directly reuses Muon’s settings (learning rate, weight decay, $\beta$). To probe robustness, we added a momentum-β sensitivity on GPT-small with LR = 6e-4 trained for 100k steps, and the training loss of the final step are shown in the table.
>
> | $\beta$       | 0.8     | 0.9     | 0.95    | 0.99    |
> | ------------- | ------- | ------- | ------- | ------- |
> | Training Loss | 2.96107 | 2.95735 | 2.95521 | 2.95487 |
>
> The resulting final losses are closely clustered, indicating stable behavior without special tuning. In practice, no extra tuning strategies were required to reproduce the reported gains. All code will be released to facilitate replication.
>
>
>
> ### **Weakness 3, second-momentum accumulation options**
>
> We have indeed compared accumulating the second moment on the raw gradient G and on the first momentum M against our default of accumulating on the orthogonalized momentum. On GPT-small, both alternatives (on G and on M) **exhibited early gradient explosion (within ≈100 steps)**, whereas accumulation on the orthogonalized direction remained stable and convergent. This justifies our design choice.
>
>
>
> ### **Question 2, MoE architecture**
>
> We trained an MoE model with a DeepSeek-V3-style architecture (≈1.3B total parameters, ≈110M activated), with the same experiment setting for Adam and AdaMuon. As show in the figure 5 in the supplementary, AdaMuon continues to perform well on MoE. These results suggest that AdaMuon adapts naturally to sparse/modular settings.

---

> > ### Comment · Reviewer_y4xw · 2025-11-27
> >
> > Thanks for the clarification and additional experiments. Although it can't scale to larger models, the experiments prove the robustness of the hyperparameters. Therefore, I tend to maintain my current score.

---

### Author Response · Authors · 2025-11-20
**General Response**

Dear AC and all reviewers:

We sincerely thank you for the time and effort you have dedicated to reviewing our work. We deeply appreciate the valuable feedback provided by the reviewers, which has allowed us to further improve the quality of our work. In response to the reviewers’ constructive comments, we have made the following revisions to our paper:

1. In response to Reviewer 8vjy weakness 2 and question 2, and Reviewer 37DQ weakness 3 and question, we have added a detailed experimental setup in Section 4.1.

2. In response to Reviewer y4xw weakness 2 and question3, and Reviewer ZYEh weakness 2, we have added an ablation study on momentum coefficient $\\beta$ in Section 4.5.

3. In response to Reviewer y4xw question 2, we have added a MoE-based experiment in Appendix D.1.

4. In response to Reviewer ZYEh weakness 1 and question, we have added more baselines for comparison in Appendix D.2.

5. In response to Reviewer ZYEh weakness 4, we have added the code repository in the supplementary material.

6. In response to Reviewer 37DQ, weakness 4, we have added a limitation section in Section 6.

We deeply value each review provided by the reviewers and sincerely appreciate how these comments have helped us enhance the quality of our paper. We hope these revisions address the concerns raised by the reviewers. Thank you again for your insightful feedback and guidance.

Sincerely,

Authors

---

### Meta-Review · Area_Chair_xxaA · 2026-01-05

**Summary:**

This paper introduces AdaMuon, an optimizer that enhances Muon by integrating element-wise second-moment adaptivity, a sign-stabilized orthogonal update, and RMS-aligned rescaling, aimed at enhancing training efficiency and stability for large-scale neural networks. This well-written and technically difficult paper tests the method on the GPT-2 and Qwen2.5 models. It often does better than AdamW and Muon when it comes to tokens-to-target and downstream performance. Most reviewers agree that the topic is important and current, that the motivation is clear, and that the empirical results show significant improvements in the conditions that were tested. The full rebuttal dealt with a lot of the problems and gaps in the research that came up during the review period. The reviewers found that the results were good, but they also found that there were major, ongoing problems with generalizability, methodological support, and experimental coverage. Taking everything into account, these issues make it very hard to recommend acceptance for now.

**Reviewer Concerns:**

Some of the issues were resolved by the rebuttal, but others still require attention:

A few experiments and some educated guesses:

Dense transformer architectures such as GPT-2 and Qwen2.5 are used in the majority of training experiments. Whether the scaling behavior holds true for larger models (more than 70 billion parameters), different architectures (such as convolutional or hybrid models), or different training schedules is still unknown.

Baseline and ablation are not adequately covered:

Usually, only AdamW and Muon are looked at. Other powerful adaptive or second-order-inspired optimizers, such as Shampoo-style optimizers or more recent large-scale ones, are not examined.

Despite the implementation of ablations, reviewers voiced concerns about the lack of comprehension of the relationships between sign stabilization, second-moment accumulation, and RMS alignment. Determining what adjustments can be made in various circumstances is difficult.

There are several distinct concepts and theories:

We still don't know if the second-moment term is useful for measuring curvature or stability, or if it's even necessary when there are more user-friendly Muon versions.

People are concerned about how easy it is to modify the training protocol and hyperparameters:

The ease of changing the learning rate, batch size, weight decay, and schedule worried reviewers.

Some reviewers believed that the robustness and fairness across optimizer-specific tuning were not sufficiently demonstrated, despite the fact that additional tests were conducted in response to the rebuttal. This made the Muon even more picky about its training.

In most cases, using real-world examples in the rebuttal improved clarity and strengthened the argument. However, the reviewers were unsure if AdaMuon offers advantages that many people can utilize, make sense in theory, and can be applied in different contexts.

**Reviewer Scores:**

Each reviewer tends to keep their original score.

---

### Decision · Program_Chairs · 2026-01-26

Reject